# Clinical and cost-effectiveness, safety and acceptability of *community intravenous antibiotic* service models: CIVAS systematic review

E D Mitchell,[1] C Czoski Murray,[1] D Meads,[2] J Minton,[3] J Wright,[2] M Twiddy[1]

▶ Prepublication history and additional material is available. To view please visit the journal (http://dx.doi.org/10.1136/bmjopen-2016-013560).

[1]Centre for Health Services Research, Leeds Institute of Health Sciences, University of Leeds, Leeds, UK
[2]Academic Unit of Health Economics, Leeds Institute of Health Sciences, University of Leeds, Leeds, UK
[3]Department of Infection and Travel Medicine, Leeds Teaching Hospitals NHS Trust, St James's Hospital, Leeds, UK

**Correspondence to**
Dr ED Mitchell;
e.d.mitchell@leeds.ac.uk

## ABSTRACT

**Objective:** Evaluate evidence of the efficacy, safety, acceptability and cost-effectiveness of outpatient parenteral antimicrobial therapy (OPAT) models.

**Design:** A systematic review.

**Data sources:** MEDLINE, EMBASE, CINAHL, Cochrane Library, National Health Service (NHS) Economic Evaluation Database (EED), Research Papers in Economics (RePEc), Tufts Cost-Effectiveness Analysis (CEA) Registry, Health Business Elite, Health Information Management Consortium (HMIC), Web of Science Proceedings, International Pharmaceutical Abstracts, British Society for Antimicrobial Chemotherapy website. Searches were undertaken from 1993 to 2015.

**Study selection:** All studies, except case reports, considering adult patients or practitioners involved in the delivery of OPAT were included. Studies combining outcomes for adults and children or non-intravenous (IV) and IV antibiotic groups were excluded, as were those focused on process of delivery or clinical effectiveness of 1 antibiotic over another. Titles/abstracts were screened by 1 reviewer (20% verified). 2 authors independently screened studies for inclusion.

**Results:** 128 studies involving >60 000 OPAT episodes were included. 22 studies (17%) did not indicate the OPAT model used; only 29 involved a comparator (23%). There was little difference in duration of OPAT treatment compared with inpatient therapy, and overall OPAT appeared to produce superior cure/improvement rates. However, when models were considered individually, outpatient delivery appeared to be less effective, and self-administration and specialist nurse delivery more effective. Drug side effects, deaths and hospital readmissions were similar to those for inpatient treatment, but there were more line-related complications. Patient satisfaction was high, with advantages seen in being able to resume daily activities and having greater freedom and control. However, most professionals perceived challenges in providing OPAT.

**Conclusions:** There were no systematic differences related to the impact of OPAT on treatment duration or adverse events. However, evidence of its clinical benefit compared with traditional inpatient treatment is lacking, primarily due to the dearth of good quality comparative studies. There was high patient satisfaction with OPAT use but the few studies considering practitioner acceptability highlighted organisational and logistic barriers to its delivery.

## Strengths and limitations of this study

- This review provides the most comprehensive picture of outpatient parenteral antimicrobial therapy (OPAT)-related research in recent years, bringing together available evidence on the clinical and cost-effectiveness, safety and acceptability of community-based antibiotic delivery.
- Much of the work in this area is based around service evaluation and many studies provided only basic descriptive findings, making it difficult to grade the quality and robustness of the evidence.
- Although many studies were identified and included in the review, the definitiveness of its conclusions related to the clinical benefit of OPAT over traditional inpatient treatment is limited by the lack of good quality comparative studies.
- Nonetheless, the synthesis provided by this review provides much needed evidence to inform the design and commissioning of OPAT services.

## INTRODUCTION

Delivery of intravenous (IV) antibiotics outside of the hospital setting (often termed outpatient parenteral antimicrobial therapy (OPAT)) is widely accepted as standard practice in many countries, including the USA and Australia.[1] In the USA alone, an estimated quarter of a million patients receive IV antibiotics on an outpatient basis each year,[2] and this method of delivery has been used to treat a wide variety of infections, among them skin and soft tissue infections, bone and joint infections, pneumonia, and endocarditis.

In recent years, OPAT services have also been developed in the UK, both in the National Health Service (NHS) and private sectors.[3] It has the potential to deliver cost efficiency savings and improved patient experience[4] through fewer inpatient bed days, antimicrobial agents that need less frequent administration and choice of treatment setting. Despite this, implementation of OPAT is patchy and there is significant geographical variation both in the setting and the healthcare professions involved. Four primary models of OPAT delivery are in use: outpatient attendance at a healthcare facility, patient (or carer) self-administration, administration by a visiting general nurse and administration by a visiting specialist nurse.

The failure to realise the full potential of OPAT in the UK is due in no small part to the paucity of information related to optimal delivery, most notably evidence of effectiveness, associated risks and patient preferences for this form of treatment. The purpose of this review was to evaluate existing evidence in relation to the efficacy, safety, acceptability and cost-effectiveness of different models of OPAT delivery.

## METHODS
### Identification of studies
A comprehensive search strategy was developed by an experienced information specialist (JW) and a range of bibliographic sources were searched for the period 1993 to March 2015: MEDLINE, MEDLINE in process, EMBASE, CINAHL, International Pharmaceutical Abstracts, the Cochrane Library, the NHS Economic Evaluation Database (EED), Research Papers in Economics (RePEc), Tufts Cost-Effectiveness Analysis (CEA) Registry and Health Business Elite (HBE). This time frame was selected to take account of the substantial changes in the organisation of the NHS and the shift towards more community involvement that have taken place following publication of the NHS and Community Care Act 1990.[5] Two separate searches were run. The first identified studies of IV antibiotics and known models of delivery, while the second identified reviews of antibiotic use in cellulitis or cystic fibrosis (infections where OPAT is frequently used) to allow for the identification of delivery models that were unknown to us and subsequently not considered when identifying terms for the first search.

Supplementary searches of Web of Science Proceedings, the Health Information Management Consortium (HMIC) and the website of the British Society for Antimicrobial Chemotherapy were conducted to identify relevant unpublished work. In addition, the reference lists of included studies were reviewed for potentially relevant papers (see online supplementary appendix 1 for sample search strategies and list of databases searched).

### Criteria for inclusion
Studies evaluating the clinical effectiveness or cost-effectiveness of an OPAT model, describing or evaluating patient safety issues associated with OPAT, or considering the acceptability of OPAT were included. The population included adult patients treated for any condition (and/or their carer in the case of acceptability studies) or practitioners involved in the delivery of OPAT. Any form of IV antibiotic drug delivery system (eg, infusion or bolus) was included. Studies of any research design were considered (with the exception of single case reports), and no language restrictions were applied.

Studies were excluded if they considered the costs related to a model of delivery but did not consider patient benefit alongside these, or if they made reference to costs and benefits but did not report specific cost-effectiveness data (such as cost per quality-adjusted life year). Similarly, studies that made reference to clinical effectiveness without reporting specific patient outcomes were also excluded. Studies that included children or that involved multiple routes of antibiotic delivery were reviewed but excluded if they did not differentiate between outcomes for adult patients or for patients receiving IV treatment, and those of other participants. Studies that focused only on the method or process of delivery or on the clinical effectiveness of a single treatment or of one class of antibiotic over another were excluded, as were abstract only, descriptive or commentary pieces and guidance documents.

### Selection of studies
Titles and abstracts of all identified studies were screened for eligibility by one reviewer (EDM) with a random selection (20%) independently screened by a second. Full-text versions of papers not excluded at this stage were obtained for detailed review and independently assessed by two reviewers (EDM with CCM, DM or MT) to determine whether they met the inclusion criteria. Differences of opinion were discussed until a consensus was reached, with validation sought from a third reviewer where necessary.

### Data extraction
Data extraction was carried out by one experienced reviewer (EDM) using a standardised pro forma. Data for a sample of studies were extracted independently by a second reviewer (CCM) in order to validate the items being collected. Information was collected on study purpose and design (including factors related to quality assessment), setting, duration, population and clinical characteristics, models of delivery (outpatient, self-administration, general nurse, specialist nurse), outcome area (clinical effectiveness, cost-effectiveness, safety, acceptability), antibiotic parameters (type, delivery route, treatment dose), outcome measures, follow-up and key findings.

### Assessment of bias
Studies were assessed for bias by two reviewers (EDM and CCM), where possible using previously developed

scoring systems. The Cochrane Risk of Bias assessment tool was used for experimental studies (randomised controlled trials, clinical trials, controlled before and after studies), and the Newcastle-Ottawa scales for cohort and case–control studies.[6 7]

A method of assessing the strength of evidence of other observational studies—developed for previous reviews on early diagnosis of cancer[8–10]—was modified for this topic area and applied to relevant studies. In this system, papers were evaluated on three key areas: population, methodology and analysis (see online supplementary appendix 2). Population assesses the method used to ensure that a study is appropriately powered/produces generalisable results, with use of a sample size calculation or inclusion of all possible patients or providers rated more highly than selective recruitment. Methodology assesses procedures for obtaining study data, with use of a rigorous approach designed to reduce systematic differences between groups (selection, characteristics, treatment, etc) rated more highly than other methods. Analysis assesses use of analytic techniques, with reporting of relevant significant comparisons or differences (or use of appropriate analytic techniques if qualitative) rated more highly than non-statistical comparisons or descriptive data.

Four categories were then used to weight the evidence: *STRONG+* (study graded strong in all three areas); *STRONG–* (study graded strong in two areas and moderate in the third); *MODERATE* (study graded strong in one area and moderate in two, or moderate in all three areas); *INSUFFICIENT* (study used a selective study population and/or an inappropriate method to ascertain data, or did not provide enough information to be able to determine a grading).

Many of the papers included in this review used methodologies that did not lend themselves to the scoring systems outlined. Many studies included all patients in receipt of OPAT since its establishment at a particular institution or over a specified time period, and simply reported conditions treated and therapies used, along with limited outcomes data. Case-series such as these, which were to all intents and purposes service evaluations that included little or no analytic content, were not subject to formal quality assessment.

In keeping with accepted good practice, studies at risk of bias were not excluded from the review, but an appraisal of the strength of existing evidence has been reported, and findings interpreted in light of this.

## Data synthesis
The main characteristics of included studies and findings relating to clinical effectiveness, cost-effectiveness, patient safety, acceptability and study quality have been summarised in narrative and tabular form. Substantial heterogeneity in clinical characteristics—condition treated, treatment duration, definition of a successful outcome (cure, improvement, deterioration, etc)—and methodology precluded pooling of study data for meta-analysis.

## Patient involvement
This review formed one work stream of a larger National Institute for Health Research (NIHR)-funded study that included qualitative interviews and economic modelling. The research team included a lay co-applicant who commented on study design prior to submission of the funding proposal, including the focus of the systematic review and the outcomes to be included. In addition, a Patient Advisory Group (PAG) was established to input at key phases throughout the project including the design of patient recruitment materials, interview topic guides and health economics aspects of the study. An Expert Panel was convened to discuss emerging findings from each work stream, and a representative of the PAG was a member of the panel. A project dissemination event was held in April 2016 to which both patients and professionals were invited.

## RESULTS
The search strategy identified 7214 articles of which 589 met the inclusion criteria for detailed review (figure 1). The full text of an additional 17 papers, identified from the reference lists of previous reviews and identified references, was also obtained giving a total of 606 potentially relevant papers. In a change to the initial protocol, non-English language papers were identified but were not assessed for inclusion (n=69), and we were unable to obtain one article from the library document supply service. One hundred and twenty-eight papers involving more than 60 000 OPAT episodes were included in the final analysis.[11–138]

## Population and setting
Three-quarters of studies were undertaken in Europe (n=53; 41%) and North America (n=45; 35%). Almost two-thirds of the European studies were UK based. Two studies involved centres in more than one country,[41 101] and one was a systematic review of world literature.[60] Studies were comparatively small in size, involving between 6 and 11 427 participants or episodes of care (mean 476; median 100); almost two-thirds (63%) had fewer than 150 participants. It was not possible to determine total participant numbers in two studies.[32 138] The period under study ranged from 6 weeks to 15 years. In general, those studies with the largest numbers of participants (>1000) either analysed all cases included in an OPAT registry and/or reviewed cases over a more substantial time period.[18 32 37 41 45 66 76 84 112 130 133]

The most commonly reported reason for treatment was osteomyelitis (n=68, 53%), followed by endocarditis (n=53, 41%), skin and soft tissue infection (n=41, 32%), cellulitis (n=32, 25%), and septic arthritis (n=29, 23%). Most studies involved multiple conditions (n=72, 56%); 7 did not specify indications for treatment. These included two qualitative studies and one survey of patient acceptability,[16 26 67] three surveys of practitioner

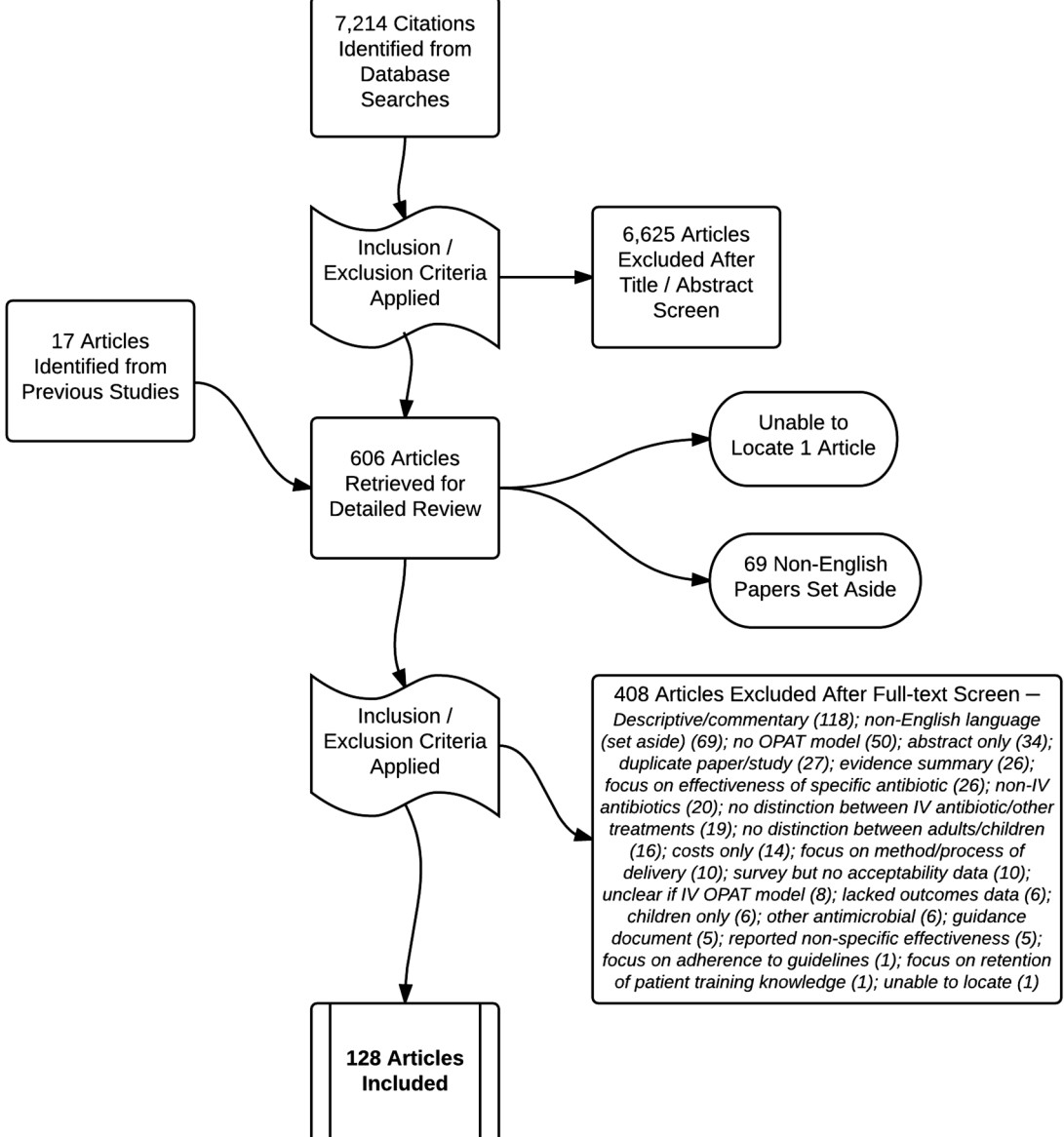

**Figure 1** Flow of studies into the review. IV, intravenous; OPAT, outpatient parenteral antimicrobial therapy.

acceptability[63 68 107] and one secondary analysis of data from an OPAT database.[17]

### Outcomes and OPAT models studied

Of the five outcome areas evaluated in the review, the most commonly considered were patient safety (n=109; 85%) and clinical effectiveness (n=89; 70%). Just over a quarter of studies involved some aspect of patient acceptability (n=37; 29%), but few determined cost-effectiveness (n=5) or practitioner acceptability (n=6). Most investigated multiple areas (70%).

Twenty-two of the 128 studies (17%) either did not indicate the type of OPAT delivery model used, or reported that home treatment was used without providing any additional detail. In the remainder, the most frequently reported model across studies was self (or carer) administration (n=66; 52%), followed by visits from a

specialist nurse (n=44; 34%), outpatient treatment (n=35; 27%) and visits from a general nurse (n=14; 11%). Just over half of these studies evaluated a single model of delivery (n=59; 55%). Other less common delivery methods and locations included infusion centres, home infusion or home care companies, hospital in the home units (HHU), prison and doctor visits (see online supplementary tables S1–S5).

### Quality assessment

Few studies employed an experimental or quasi-experimental design (see online supplementary tables S1–S5). Most (67%) were observational, primarily case-series, and many involved retrospective data collection (n=57; 45%). Of the 12 included randomised controlled trials, 5 reported on subgroup analyses from the main study and all failed to provide details of the trial

methodology.[15 77 99 100 102] One of two included clinical trials did not specify the delivery model under evaluation.[39]

Many studies analysed data obtained from medical records or prospectively held OPAT databases (42%); a small number of others carried out secondary analysis of national or international OPAT registries. Satisfaction surveys, either by questionnaire or telephone completion, were frequently used, while interviews (face-to-face, telephone, focus group) and visual analogue scales were also well represented. Less common methods of data collection included clinic or ward diaries, direct observation, and data from previous studies or the published literature. It was often unclear who collected data, and few studies reported on how this was carried out. A substantial number provided little or no detail on the methods used. Twenty-one of the 128 (16%) studies received full or partial funding by pharmaceutical companies, while 72 others (56%) did not report the funding source.

### Risk of bias within studies

Three of the 14 included trials were assessed as having a low risk of bias,[30 103 131] and 1 a high risk of bias (pilot study where patients self-selected hospital or home treatment and were recruited consecutively to each arm after this decision was made).[40] In the remaining 10 studies, the level of potential bias was unclear. In five cases, it was uncertain whether randomisation or controlling had taken place. Four of these studies reported on subgroup analyses from a single open-label trial, and none provided details of the original study methodology.[15 77 99 100] The fifth reported on two related trials comparing IV with oral treatment for neutropenia in patients with cancer, and again reported no methodological details of the parent studies.[39] In the remaining five trials, details of randomisation, allocation concealment and blinding (especially in relation to assessment of outcome measures) were poorly reported.[101 102 111 116 132] In addition, one of the trials closed early due to poor recruitment,[87] while population bias due to age and treatment duration differences could not be excluded from another.[132]

All five included case–control studies (three of them retrospective) rated low for potential bias (median 8/9; range 6–9).[44 78 98 136 137] Those studies scoring lowest did not provide details on the methods used to determine outcomes.[44 78] Similarly, four cohort studies (two retrospective) all had low potential for bias (median 8/9; range 7–9);[12 20 76 112] lower scores related to possible failure to control for potential differences.[20 76]

In the majority of observational studies, the data analysed were derived from OPAT databases or medical records review. Many studies also involved questionnaire surveys. Only five included a comparator (inpatient care),[23 61 75 81 122] with many simply including all OPAT patients over a selected time period. Most studies reported descriptive results only, with no statistical testing of differences.

### Impact on clinical outcomes

Only 21 of the 89 studies evaluating the clinical effectiveness of OPAT included a comparator which, with few exceptions, was inpatient treatment (table 1; see online supplementary table S1). Five of the studies did not specify the OPAT delivery model used,[39 75 122 132 136] while two others reported combined results for multiple models.[102 137]

Regardless of the particular OPAT model used, most studies assessing duration of treatment found little difference compared with inpatient therapy[30 32 40 44 81 98 101 103 131] (table 1). Two studies (one using specialist nurses,[18] the other self-administration[74]) found that the average length of OPAT treatment was shorter. A third study evaluating a HHU found a significant reduction in treatment duration for patients with cellulitis (median 6 vs 8.6 days; 95% CI 0.24 to 4.85) but not for patients with pyelonephritis (4.6 vs 4 days; 95% CI −1.86 to 0.72).[81]

Evidence of the effect of OPAT on cure rate, however, was less conclusive. When findings from all delivery models were considered, OPAT appeared to produce superior results to those seen for inpatient treatment. This may reflect heterogeneity in the population groups, with inpatients being thus because they required more intensive care. However, it is also undoubtedly influenced by the inclusion of positive studies that reported on multiple or unspecified OPAT treatment models.[75 102 137] When these were removed and specific models considered individually, outpatient delivery appeared to have a lower rate of cure or improvement than inpatient treatment (89% vs 95%)[101] and self-administration[20] and OPAT by a specialist nurse a higher rate.[44 81] One study found that duration of neutropenia-related fever in patients with cancer was significantly reduced with specialist nurse-delivered OPAT (2 vs 5 days; p<0.001),[44] and another that rates of recovery for pyelonephritis were higher (93% vs 90%),[62] although this was not the case for cellulitis (93% vs 100%).[81] OPAT delivered by a general nurse, however, had little or no impact on rates of recovery. One study of treatment for community-acquired pneumonia[103] found no significant difference in time to resolution of fever, tachycardia and tachypnoea, while another found no significant difference in days to no advancement of cellulitis[30] (mean 1.50 vs 1.49; 95% CI −0.3 to 0.28; p=0.90). Results from studies assessing the impact of OPAT, specifically on lung function in patients with cystic fibrosis, were either inconclusive or showed no differential impact.[23 40 98 131]

In the remaining studies that considered clinical effectiveness for OPAT models only, rates of cure or improvement ranged from 61.1% to 100% (mean 89.6%; median 92.5%). When the various OPAT models were considered individually, the highest mean cure/improvement rate was found for self-administration (91.3%), followed by specialist nurse (90.6%), general nurse (90.0%) and outpatient

**Table 1** Effect of OPAT on clinical success and safety

| OPAT model(s) [Study ID] | Inferior | Superior | No difference | Inconclusive |
|---|---|---|---|---|
| **Duration of treatment (all models)** | | | ◉ | |
| Outpatient attendance vs. inpatient treatment [101] | | | ● | |
| Self-administration vs. inpatient treatment [40, 98, 131] | | | ● | |
| General nurse vs. inpatient treatment [30, 103] | | | ● | |
| Specialist nurse vs. inpatient treatment [32, 44, 81] | | | ◉ | |
| **Rate of cure or improvement (all models)** | | ◉ | | |
| Outpatient attendance vs. inpatient treatment [101] | ● | | | |
| Self-administration vs. inpatient treatment [20] | | ● | | |
| General nurse vs. inpatient treatment [30,103] | | | ● | |
| Specialist nurse vs. inpatient treatment [44, 81] | ◉ | | | |
| **Improved lung function in CF (all models)** | | | | ○ |
| Self-administration vs. inpatient treatment for $FEV_1$ [23, 40, 98, 131] | | | ● | |
| Self-administration vs. inpatient treatment for FVC [23, 40, 98, 131] | | | | ○ |
| Self-administration vs. inpatient treatment for PEFR [23, 98] | | | | ○ |
| **Drug-related side effects (all models)** | | | ◉ | |
| Outpatient attendance vs. inpatient treatment [101] | | ● | | |
| Self-administration vs. inpatient treatment [20, 40, 98, 131] | | | ● | |
| General nurse vs. inpatient treatment [103] | | | ● | |
| Specialist nurse vs. inpatient treatment [44, 78, 81, 109] | | | | ○ |
| **Venous access complications (all models)** | ◉ | | | |
| Self-administration vs. inpatient treatment [20, 40, 131] | | | ◉ | |
| General nurse vs. inpatient treatment [103] | ● | | | |
| **Hospital admission (all models)** | | | | ○ |
| Self-administration vs. inpatient treatment [20, 131] | | | ● | |
| General nurse vs. inpatient treatment [30, 103] | ● | | | |
| Specialist nurse vs. inpatient treatment [32, 78, 81, 109] | | ◉ | | |
| **Deaths (all models)** | | | ● | |
| Outpatient attendance vs. inpatient treatment [101] | ● | | | |
| Self-administration vs. inpatient treatment [20, 131] | | | ● | |
| General nurse vs. inpatient treatment [103] | | | ● | |
| Specialist nurse vs. inpatient treatment [32, 44] | | | ● | |

■ (dark) Reports the overall effect on an outcome for all OPAT models combined. This includes findings from studies where the OPAT model was not specified or where results for multiple models were not reported separately.

□ (light) Reports the effect only for studies that specify the model under study and report on individual findings.

Key: ● found in ≥75% studies considering outcome; ◉ found in ≥50% studies considering outcome; ○ evidence of effect supported by <50% studies considering outcome.

CF, cystic fibrosis; $FEV_1$, forced expiratory volume in 1 s; FVC, forced vital capacity; OPAT, outpatient parenteral antimicrobial therapy; PEFR, peak expiratory flow rate.

treatment (88.3%). Few studies reported on bacterial eradication, but those that did saw rates of between 57.1% and 100% (mean 86.2%; median 90.0%; see online supplementary table S1).

### Patient safety and adverse events

Only 24 of the 109 studies evaluating OPAT-related safety included a comparator, which in the majority of cases was inpatient treatment (table 1; see online supplementary table S2). Five studies did not specify the OPAT model being used[39 61 75 132 136] and three others reported combined results for multiple models.[76 102 137]

There was little evidence of impact on either drug-related side effects or number of deaths in OPAT patients compared with patients receiving treatment in hospital[20 32 40 44 76 81 98 101 103 109 131] (table 1). One study evaluating OPAT via outpatient treatment[101] reported a higher death rate (1 vs 0 patients), but this was a small study and the overall rate of side effects was lower in the OPAT group (15% vs 18%). There also appeared to be no conclusive evidence of benefit either in relation to hospital readmissions overall or in relation to those who self-administered therapy.[20 131] There were conflicting results for OPAT delivered by nurses, with specialist nurses[32 78 81 109] appearing to have superior results to those of general nurses.[30 103] Perhaps unsurprisingly, overall there appeared to be more line-related complications in IV therapy administered outside of hospital.

Across all studies, the most commonly reported adverse events were rash, fever, nausea/vomiting, diarrhoea, allergic reaction or anaphylaxis, phlebitis, leucopenia and line complications (including line infection, occlusion, breakage and dislodgement; see online supplementary table S2).

### Cost-effectiveness of OPAT

While many of the identified studies reported on the cost of OPAT, only five considered cost-effectiveness (see online supplementary table S3). Three studies applied decision tree models to OPAT provided by specialist nurses,[95 119 138] with one also determining the cost-effectiveness of self-administration.[138] The remaining studies (one literature review, one retrospective observational study) did not specify the OPAT model(s) used.[60 122]

In two of the three decision tree analyses, IV OPAT was found to be more cost-effective than IV inpatient therapy.[119 138] In one case, it was also more cost-effective than early discharge with oral therapy and oral outpatient therapy.[119] In the other, its dominance was only maintained when the IV success rate was >55%.[138] Conversely, in the third study, IV OPAT was found to be less cost-effective than IV to oral switch therapy and oral treatment, both during and after hospitalisation (which was the most cost-effective option).[95] However, the probability estimates used in two of the three scenarios in this study were derived from different data sources (published research and institutional data). In addition,

there were no published data related to the baseline probability for cure in the third scenario, and so this was assumed to be at some point between the estimates used in the other two scenarios.

Studies included in the literature review predominantly concluded that home IV antibacterial therapy would lead to significant cost reductions, both from a societal and third-party payer perspective.[60] In 5 of the 11 studies, inpatient therapy was 2–3 times as expensive as home care therapy. However, there was considerable variation in the way in which costs were determined and calculated across studies (eg, incremental costs, costs for selected components only, etc), while the review itself lacked considerable detail on the methodology used and the criteria for study inclusion and exclusion.

Findings from the observational study (based on 435 courses of IV antibiotic treatment for respiratory exacerbations in 116 adult patients with cystic fibrosis) indicated that for both 1 course and 1 year of treatment, IV antibiotic treatment administered mostly in hospital was more effective but more costly than treatment administered mostly at home.[122] This improved clinical effectiveness could only be achieved with the input of considerable additional resources (between £46 and £73k per patient at 2002 prices). However, when the strictest definition of effectiveness was applied (≤0% decline in lung function), hospital treatment was unlikely to ever be cost-effective.

### Acceptability of OPAT
#### Patient preference

Only 4 of the 36 studies considering patients' acceptability of OPAT involved a comparison of inpatient and outpatient therapy (see online supplementary table S4), 2 where OPAT delivery was by general nurses[30 103] and 2 where delivery was by specialist nurses.[32 44] In each case, satisfaction was high, with home treatment seen as beneficial. One of the two general nurse studies found that only 5% of home group patients would have preferred hospital treatment, compared with 35% of hospital patients who would have preferred treatment at home (p<0.001).[30] The other found that patients in the home group were significantly happier with the location of their care than those receiving inpatient therapy (p<0.001).[103] Similarly, in one of the studies where patients received home care by specialist nurses, almost all (97%) indicated that they would choose to receive at-home therapy in future and would recommend it to others.[44] The main reasons given for this were quiet and increased home comfort, familiar environment, and free choice of activity. However, some patients in this study did report disadvantages to receiving therapy at home, primarily related to patient and caregiver anxiety.[44]

In the remaining studies that considered acceptability in OPAT patients only, most involved multiple OPAT models (and did not differentiate between them in their findings), or did not specify the model(s) under study. For the most part, satisfaction with treatment was very

high,[11 26 28 42 45 48 51 52 55–57 59 74 89 90 104 105 108 113 120 123 124] including where patients had to have frequent attendance at hospital.[22 135] Commonly perceived advantages of OPAT included the ability to resume daily activities,[45 56 59 74] feelings of improved self-esteem or greater freedom and control,[48 56 74 82 93] and not having to remain in or attend hospital.[11 56 59 82 118] The main disadvantages were most commonly related to infusion equipment, and included anxiety about the device and its sterility.[16 67 97] Two studies found that younger patients were better able to use infusion devices, and required less support to do so than older patients.[31 67]

Two studies determined patients' willingness to pay to have treatment in their preferred location, and although differences did not reach statistical significance, patients reported that they would pay more for the home-based than hospital-based treatment, including giving up slightly more of their remaining life to ensure this.[73 120]

### Provider preference

Only six studies included assessment of practitioner acceptability: one involving general practitioners (GPs), one involving nurses and four involving infection specialists (see online supplementary table S5). In most cases, professionals saw advantages (or a need) for patients receiving IV antibiotic therapy outside of hospital.[68 86 93 107] However, there were also negative perceptions in relation to practitioner involvement. Most GPs saw no advantage to themselves in home treatment, and many thought distance from hospital was an issue for patients.[93] Similarly, nurses perceived that there were challenges in providing this model of care, mainly around the technical nature of the devices used, and dealing with patients' understanding of the technology and its related risks.[68] Finally, many specialists saw organisational barriers to the use of OPAT, in terms of funding, leadership, and the links between primary and secondary care.[63 86–87 107] In addition, there was no consensus regarding who should assume the cost or take clinical responsibility for patients.[87 107]

## DISCUSSION
### Principal findings

Rates of cure from the published literature, when aggregated, show that there are no systematic differences in relation to the impact of OPAT on duration of therapy, or on adverse events associated with IV antibiotic treatment. In addition, OPAT is, on the whole, more cost-effective than inpatient care. However, conclusive evidence as to the clinical benefit (or otherwise) of this mode of therapy compared with traditional inpatient IV treatment is lacking. Acceptability of OPAT appears to be high among patients, who appreciate the greater freedom that it provides, particularly in relation to being able to resume daily activities (such as going to work or school), having greater control over their illness and not having to attend hospital but being able to stay at home with family. Disadvantages identified most commonly

related to the use of infusion equipment. Few studies considered practitioner acceptability, but those that did found some concerns related to the logistics involved in providing an OPAT service, including cost and who would assume clinical responsibility for patients.

The results of this review should be used by clinicians in the context of their patient populations and the types of infection being treated. Results from ongoing research studies, such as work being done by Scarborough et al[139] to investigate the possibility of early switch from inpatient IV to oral antibiotics, are likely to change the way in which patient care is planned and implemented. However, the results of this review (within the constraints of the literature) have demonstrated that OPAT is safe, effective and cost-effective and, as such, patients are likely to continue to be treated using OPAT models. In addition, the evidence shows that patients prefer OPAT, so it is likely that there will be continued development of community-based services.

### Limitations of the study

Although many studies were identified and included in this review, its conclusions are limited by the lack of studies involving a usual care comparison, or comparison with other models of OPAT delivery. In addition, few studies employed a rigorous study design. Much of the work in this area appears to be based around service evaluation and, as such, many of the studies provided only basic descriptive findings, with no estimates of variance and limited data related to patient outcomes. As a result, it is difficult to grade the quality and robustness of the evidence, even in the few randomised controlled trials that have been conducted. Similarly, substantial clinical and methodological heterogeneity precluded pooling of data for meta-analysis. The majority of studies included a varied case-mix and did not differentiate their results between conditions treated. In addition, there was variation across studies in relation to what constituted a successful outcome (cure, improvement, deterioration, etc), as well as a lack of consistent treatment duration. Although this undoubtedly reflects real-world practice, it meant that it was not possible to pool the results of individual studies to provide estimates of true effect size when using OPAT for different patient groups, or even for comparing OPAT as a whole with inpatient treatment. Although identified as part of the search strategy, time constraints meant that non-English language papers were not assessed beyond the abstract screening stage, and it is possible that this may have resulted in some language bias (through exclusion of negative findings that may be more likely to have been published in non-international journals). However, given the lack of robust, conclusive studies that were included, this is unlikely to have significantly influenced the findings of the review.

### Implications for clinicians and policymakers

It is likely that the increased use of OPAT in the UK in recent years is due, in no small part, to clinicians'

expectations that it should increase hospital inpatient capacity while delivering non-inferior patient care and improved patient experience. However, this review has demonstrated that the evidence related to care is not strong. Few studies reported on the different levels of service required to account for the complexity of patient cases encountered, including those with comorbidity and those requiring 'one off' or longer term treatment. In addition, many studies provided aggregated results and it was not possible to disentangle these either for individual OPAT models or for the specific conditions treated. This, combined with a lack of detail on the delivery models actually used, makes it difficult for clinicians and policymakers to be able to replicate existing practice (and, consequently, potential outcomes) even from positive studies. OPAT services have the potential to deliver significant cost savings and increased patient satisfaction for the NHS, and it is crucial that this information is reported in future studies in order to identify best practice and help support decision-making at a local level. We would recommend, that as a minimum, authors fully describe the specific method of OPAT delivery studied (including defining who delivers 'specialist' nurse care), include and report on a comparator when determining the impact of OPAT on clinical outcomes (only 21/89 studies in this review did so), report all numerators and denominators, avoid reporting pooled data (where possible), report on case complexity and average duration of treatment to achieve cure, and adhere to CONSORT guidelines when reporting trials.

## Future research

Commissioning healthcare services in England is increasingly complex and has been subject to much change in recent years.[140] The NHS England service specification for infectious diseases services[141] includes a requirement to provide OPAT, but at present there is no clear commissioning mechanism to achieve this. This review has established that there is currently a dearth of information around key aspects of OPAT, and it is therefore clear that where Clinical Commissioning Groups have opted to commission OPAT services, they are working without the support of a robust evidence base. It is also clear that there is little evidence on which recommendations for best practice can be based. Indeed, the main guidance currently in use is the British Society for Antimicrobial Chemotherapy OPAT standards document,[142] which is a consensus statement largely derived from expert opinion.

There is a need for further research to address significant gaps in knowledge about the cost-effectiveness of different IV antibiotic services, and to ascertain which services patients prefer and which aspects of the services are most important to them. This would help identify the most appropriate configuration of services in terms of value for money and patient preference, as well as future research priorities. Since OPAT services are now established in many areas of the UK, conducting a definite randomised controlled trial of different models of care is unlikely to be feasible. Individual centres would not be able to provide all of the necessary comparator services for randomisation, and there would be difficulties related to the heterogeneous nature of the patient population, and consequent decisions about the most relevant outcome measure. Other, more pragmatic, study designs should therefore be considered, including large-scale, prospective observational studies designed to collect robust data on outcomes, adverse events and the costs of OPAT (including mixed models of care). This, together with high standards of reporting to ensure that delivery models and their results are fully described and reported individually, would go some way to bridging the current information gap.

**Acknowledgements** The authors would like to thank Thomas Veale for help in obtaining papers, Jill Edwards for help with screening abstracts, and Kate Hill for assistance with data extraction. The authors would also like to thank the members of the CIVAS Patient Advisory Group and Expert Panel.

**Contributors** The review was designed by CCM, JM, JW and MT as part of a larger study. JW developed and ran the electronic searches. Data extraction tools were designed by EDM. EDM, CCM, DM and MT were involved in record screening and study selection. EDM and CCM undertook data abstraction and carried out assessment of risk of bias. EDM synthesised the data. The article was drafted by EDM, and all authors contributed to subsequent drafts. EDM is the guarantor.

**Funding** This work was supported by NIHR Health Services Research and Delivery (HSDR), Project reference 11/2003/60.

**Competing interests** None declared.

**Provenance and peer review** Not commissioned; externally peer reviewed.

**Data sharing statement** No additional data are available.

**Review registration** This study is registered as PROSPERO CRD42013006374 (available at http://www.crd.york.ac.uk/PROSPERO/).

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
