## [Reviewer comments · BMJ Open]

ARTICLE DETAILS

TITLE (PROVISIONAL)	Clinical and cost-effectiveness, safety and acceptability of community intravenous antibiotic service models: CIVAS systematic review
AUTHORS	Mitchell, Elizabeth; Czoski Murray, Carolyn; Meads, David; Minton, Jane; Wright, Judy; Twiddy, Maureen

VERSION 1 - REVIEW

REVIEWER	João Gonçalves-Pereira ICU Hospital Vila Franca de Xira Portugal
REVIEW RETURNED	10-Oct-2016

GENERAL COMMENTS	Mitchell et al. provide an extensive review on published data regarding the use of Outpatient Parenteral Antibiotic Therapy (OPAT). Overall they identified 128 studies including more than 60.000 episodes of OPAT undertaken between 1993 and 2015. These studies were largely heterogeneous and mostly related to service evaluation and proof of feasibility. Nevertheless the vast majority of studies (including those 29 with a comparator group) showed that OPAT was safe, effective. Studies addressing costs and patient satisfaction showed a benefit of OPAT. However professionals perceived several challenges related to its practice. The authors weren't able to provide any pooling of data to the significant heterogeneity. Consequently they conclude that there is a need for good quality, comparative studies (OPAT vs. inpatient therapy). This study provides a comprehensive review of OPAT, bringing together the available published evidence. There are some details, mainly related to the presentation of data, which I think authors should address: 1- I think the manuscript is too descriptive. Authors repeat information that is already in the tables (provided as supplementary material). I think a more interpretative summary will be helpful for the reader. 2- Table 1 can be summarized in figure 1. 3- Table 2 can be removed. Again most of the important information is provided as supplementary material. I think authors can provide instead a short list of the number of studies addressing each different question. 4- In table 3, although I agree that it is difficult to pool the data (as the
--

	studies are heterogeneous), authors could provide, when superiority or inferiority is present, the calculated differences in each study. 5- Finally, I think authors should provide some recommendation for minimum data that should be present in future research.
--	---

REVIEWER	Eavan Muldoon University Hospital South Manchester, UK
REVIEW RETURNED	26-Oct-2016

GENERAL COMMENTS	This is an ambitious paper, and I commend the authors for their efforts. While the results are unsurprising, I think it is well worthy of publication, as it highlights the paucity and deficiencies in the OPAT literature. I have one issue with the paper. In the discussion, the authors talk about care which is better than hospital care, however surely the focus should be on care that is non inferior to hospital care?
--

REVIEWER	Jørgen T Lauridsen University of Southern Denmark
REVIEW RETURNED	06-Dec-2016

GENERAL COMMENTS	The study presents a systematic review of literature on clinical and cost effectiveness, safety and acceptability of CIVAS models, which has not been performed previously. It is professionally performed and thus lives up to my requests. Therefore, I recommend accept.
--

VERSION 1 – AUTHOR RESPONSE

Reviewer: 1 (João Gonçalves-Pereira)

1. "I think the manuscript is too descriptive. Authors repeat information that is already in the tables (provided as supplementary material). I think a more interpretative summary will be helpful for the reader

It is usual for the Results section of a systematic review to only describe findings, and for interpretation to be provided in the Discussion section. We believe that supplementing the tables in this way is necessary to make the synthesised findings easily accessible to readers. We have reworded and added additional text to the "Principal findings" section of the Discussion to provide further interpretation (Page 14).

2. Table 1 can be summarised in figure 1

We have incorporated reasons for study exclusion into Figure 1 as suggested, and have removed Table 1 along with reference to it on Page 7 of the text.

3. Table 2 can be removed. Again most of the important information is provided as supplementary material. I think authors can provide instead a short list of the number of studies addressing each different question

Details on the number of studies addressing each question is already presented in the Outcomes and

OPAT models studies section of the Results (Page 8). We have therefore simply removed Table 2 along with reference to it on Page 7 of the text.

4. In table 3, although I agree that is difficult to pool the data (as the studies are heterogeneous), authors could provide, when superiority or inferiority is present, the calculated differences in each study

We thank the reviewer for this suggestion. However, this applies only to a minority of the entries in Table 3 (now Table 1), and most of the relevant information is provided in the text of the Results section. We have therefore not amended the table but will be happy to do so if the editorial team feel that this is necessary.

5. Finally, I think authors should provide some recommendation for minimum data that should be present in future research

We thank the reviewer for this useful suggestion, and have added a section to the Discussion to address this point (Pages 15-16).

Reviewer: 2 (Eavan Muldoon)

1. In the discussion, the authors talk about care which is better than hospital care, however surely the focus should be on care that is non inferior to hospital care?

We thank the reviewer for highlighting this point, and have revised the text of the Discussion accordingly (Page 15).

VERSION 2 – REVIEW

REVIEWER	João Gonçalves-Pereira Hospital Vila Franca de Xira Portugal
REVIEW RETURNED	28-Jan-2017

GENERAL COMMENTS	I found the manuscript much more easy to read. I have no further comments
--